ecology, environmental science

pollution, seabirds, *Thalassarche chrysostoma*, heavy metals, Southern Ocean

**Author for correspondence:**
William F. Mills
e-mail: wilmil23@bas.ac.uk

# Mercury exposure in an endangered seabird: long-term changes and relationships with trophic ecology and breeding success

William F. Mills[1,2], Paco Bustamante[3,4], Rona A. R. McGill[5], Orea R. J. Anderson[6], Stuart Bearhop[2], Yves Cherel[7], Stephen C. Votier[8] and Richard A. Phillips[1]

[1]British Antarctic Survey, Natural Environment Research Council, Cambridge CB3 0ET, UK
[2]Centre for Ecology and Conservation, University of Exeter, Cornwall TR10 9EZ, UK
[3]Littoral Environnement et Sociétés (LIENSs), UMR 7266, CNRS-La Rochelle Université, 2 rue Olympe de Gouges, 17000 La Rochelle, France
[4]Institut Universitaire de France (IUF), 1 rue Descartes, 75005 Paris, France
[5]NERC Life Sciences Mass Spectrometry Facility, Scottish Universities Environmental Research Centre, East Kilbride G75 0QF, UK
[6]Joint Nature Conservation Committee, Inverdee House, Baxter Street, Aberdeen AB11 9QA, UK
[7]Centre d'Etudes Biologiques de Chizé (CEBC), UMR 7372 du CNRS-La Rochelle Université, 79360 Villiers-en-Bois, France
[8]Lyell Centre, Heriot-Watt University, Edinburgh, UK

WFM, 0000-0001-7170-5794; PB, 0000-0003-3877-9390; RARM, 0000-0003-0400-7288; ORJA, 0000-0002-9672-7313; SB, 0000-0002-5864-0129; YC, 0000-0001-9469-9489; SCV, 0000-0002-0976-0167

Mercury (Hg) is an environmental contaminant which, at high concentrations, can negatively influence avian physiology and demography. Albatrosses (Diomedeidae) have higher Hg burdens than all other avian families. Here, we measure total Hg (THg) concentrations of body feathers from adult grey-headed albatrosses (*Thalassarche chrysostoma*) at South Georgia. Specifically, we (i) analyse temporal trends at South Georgia (1989–2013) and make comparisons with other breeding populations; (ii) identify factors driving variation in THg concentrations and (iii) examine relationships with breeding success. Mean ± s.d. feather THg concentrations were $13.0 \pm 8.0 \, \mu g \, g^{-1}$ dw, which represents a threefold increase over the past 25 years at South Georgia and is the highest recorded in the *Thalassarche* genus. Foraging habitat, inferred from stable isotope ratios of carbon ($\delta^{13}$C), significantly influenced THg concentrations—feathers moulted in Antarctic waters had far lower THg concentrations than those moulted in subantarctic or subtropical waters. THg concentrations also increased with trophic level ($\delta^{15}$N), reflecting the biomagnification process. There was limited support for the influence of sex, age and previous breeding outcome on feather THg concentrations. However, in males, Hg exposure was correlated with breeding outcome—failed birds had significantly higher feather THg concentrations than successful birds. These results provide key insights into the drivers and consequences of Hg exposure in this globally important albatross population.

## 1. Introduction

Mercury (Hg) is a pervasive environmental contaminant that can negatively impact humans and wildlife [1]. Hg derives from both natural and anthropogenic sources; however, human activities have increased the global Hg pool and projections suggest that global anthropogenic Hg emissions are likely to increase [2–4].

In its gaseous, elemental form, Hg can travel long distances to remote locations through atmospheric transport [5]. Once deposited in the marine environment, inorganic Hg (iHg) is converted (through methylation) to the more toxic form, methyl-Hg ($[CH_3Hg]^+$; or MeHg), which, once assimilated, bioaccumulates in marine organisms and biomagnifies through food webs from lower to higher trophic levels [6,7]. Long-lived marine top predators, such as many seabird species, are therefore potentially exposed to high Hg concentrations through their prey [8].

Seabirds are often used as indicators of marine ecosystem health [9], including the bioavailability of Hg [10,11]. Studies of seabird communities in the Southern Hemisphere, from Antarctica to the subtropics, have revealed considerable interspecific variation in Hg contamination [12–17]. The Procellariiformes (albatrosses and petrels) are the most abundant and diverse seabird group in the Southern Ocean, and albatrosses (Diomedeidae) are consistently the most contaminated family of birds [18–20]. Phylogeny exerts a major influence on Hg exposure among albatrosses, such that members of the *Diomedea* genus typically display the highest levels [12,18]. Indeed, in the oceans, only some marine mammals show higher Hg concentrations [21]. Hg contamination within seabird populations is often highly variable and governed by factors such as age, sex, breeding status and foraging ecology [8,16,22]. Moreover, at high concentrations, Hg can have fitness consequences [23–25]. To date, the drivers of intraspecific variation in Hg contamination, and its consequences for breeding and survival, have only been studied in one albatross species, the wandering albatross (*Diomedea exulans*) [26–29].

The present study focuses on grey-headed albatrosses (*Thalassarche chrysostoma*) breeding at South Georgia—a remote island south of the Antarctic Polar Front (APF) in the Atlantic sector of the Southern Ocean. South Georgia hosted 47 674 breeding pairs of this species in 2003/2004, which represented by far the largest population globally (approx. 50%), but has been in long-term decline and is currently listed as endangered by the World Conservation Union [30,31]. This species is extremely long-lived [32], and forages at mid- to high trophic levels—predominantly consuming cephalopods, but also fish and Antarctic krill (*Euphausia superba*) [33,34]. During the non-breeding period, free from the constraints of central-place foraging, birds disperse across a wide range of oceanic habitats though mainly targeting the Subantarctic Zone (SAZ) between the APF and the Subtropical Front (STF) [35–37]. Hg exposure tends to be high in this species, with intrapopulation variation reflecting its generalist feeding habits and wide foraging range [18]. In this cross-sectional study, we measured feather THg concentrations in a very large sample of individuals of known age, sex and breeding history. The vast majority (>90%) of the THg excreted into albatross body feathers is MeHg [18,19,38]. Our main aims were to: (i) analyse long-term trends in Hg exposure of this species at South Georgia (1989–2014); (ii) examine large-scale spatial variation in Hg exposure by comparing our data with other breeding populations [12,14,18,39]; (iii) assess Hg exposure in relation to intrinsic factors (sex, age and breeding history) and, using stable isotope ratios as proxies, to foraging habitat ($\delta^{13}C$) and trophic level ($\delta^{15}N$) and (iv) test for short-term relationships between Hg exposure and subsequent breeding outcome. This final point is of particular interest, given that breeding success in this population is low and highly variable, contributing to its long-term decline [40].

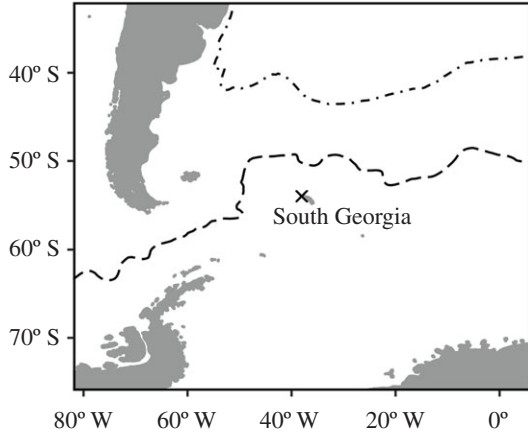

**Figure 1.** Location of Bird Island, South Georgia (cross), in relation to the APF (dashed line) and STF (dot-dash line).

## 2. Material and methods

### (a) Study site and sample collection

Fieldwork was carried out at Bird Island, South Georgia (54°00′ S, 38°03′ W) (figure 1). Since the 1970s, intensive study-colonies of grey-headed albatrosses have been visited daily to weekly throughout the breeding season to record the identities of breeders and non-breeders, laying and fledging dates, and nest survival [40]. Chicks have been ringed annually and the modal age of first breeding is 12 years [41]. Birds were sexed from records of observed copulatory position, egg-laying attendance pattern or using DNA extracted from blood samples [42]. All birds bred in the sampling year. A random collection of relatively unabraded body feathers were obtained from the breast region of adults ($n = 78$) of known breeding history in December and January in the 2013/2014 breeding season. Grey-headed albatrosses are not in active body feather moult at Bird Island between October and February [43]; hence, THg concentrations and stable isotope ratios (see below) of body feathers sampled during the breeding season should reflect Hg burdens, foraging areas and trophic levels when grown in the preceding non-breeding period. Previous studies of Hg in grey-headed albatrosses at South Georgia (reflecting THg concentrations in 1989, 1998 and 2001) were of feathers collected in a similar manner.

### (b) Total mercury analysis

Hg in feathers is essentially inert and cannot be reincorporated into body tissues [44]. Feathers were cleaned of surface lipids and contaminants using chloroform : methanol solution (2 : 1v/v) followed by successive Milli-Q® water rinses. Feathers were air-dried and cut into small fragments with stainless steel scissors. For each individual, three feathers were analysed separately and whole feathers were analysed excluding only the rachis. Feather THg concentrations were measured using an Advanced Mercury Analyser spectrophotometer (Altec AMA 254) at the laboratory Littoral Environnement et Sociétés (LIENSs, France). For each individual feather, a minimum of two aliquots (range: 0.40–1.50 mg dry weight [dw]) were analysed, and the means and relative standard deviation between measurements were calculated (all samples RSD <10%). THg concentrations are presented in $\mu g\ g^{-1}$ dw. Accuracy was tested using a certified reference material (dogfish liver DOLT-5, NRC, Canada; certified Hg concentration: $0.44 \pm 0.18\ \mu g\ g^{-1}$ dw) every 10 samples. The measured values were $0.46 \pm 0.02\ \mu g\ g^{-1}$ dw ($n = 21$), and thus, the recovery rate was $105 \pm 5\%$. Blanks were analysed at the beginning of each set of samples and the detection limit of the method was $0.005\ \mu g\ g^{-1}$ dw.

**Table 1.** Results from previous studies measuring total mercury (THg) concentrations (µg g$^{-1}$ dw) in body feathers of grey-headed albatrosses (*T. chrysostoma*). Values presented are means ± s.d. (range). Sampling procedure refers to whether THg concentrations were measured separately in multiple feathers (individual) or if multiple feathers were pooled before analysis (pooled).

| breeding site location | sampling year | n | sampling procedure | feather THg (µg g$^{-1}$ dw) | reference |
|---|---|---|---|---|---|
| South Georgia | 1989 | 34 | pooled | 4.20 ± 2.27 (1.22–11.00) | Thompson *et al.* [39] |
| | 1998 | 19 | pooled | 8.93 ± 2.85 | Becker *et al.* [14] |
| | 2002 | 15 | pooled | 9.50 ± 2.84 (4.34–13.24) | Anderson *et al.* [12] |
| | 2006 | 10 | individual[a] | 7.35 ± 7.57 (2.12–28.25) | Cherel *et al.* [18] |
| | 2014 | 78 | pooled | 13.08 ± 6.56 (3.46–31.65) | present study |
| | 2014 | 229 | individual | 13.04 ± 8.03 (2.06–35.14) | present study |
| Campbell Island | 1988 | 30 | pooled | 6.91 ± 2.40 (3.10–13.63) | Thompson *et al.* [39] |
| | 2013 | 20 | individual[a] | 9.50 ± 3.11 (3.79–15.53) | Cherel *et al.* [18] |
| Prince Edward Islands | 2006 | 11 | individual[a] | 7.12 ± 3.21 (3.00–14.07) | Cherel *et al.* [18] |

[a]Analyses were based on a single feather per individual.

## (c) Stable isotope analysis

Stable isotope ratios of carbon ($^{13}$C/$^{12}$C, expressed as $\delta^{13}$C) and nitrogen ($^{15}$N/$^{14}$N, $\delta^{15}$N) were measured on the same individual feathers as THg concentrations. Feather stable isotope ratios reflect those of prey during the period of their synthesis, and because they are metabolically inert once grown they preserve an isotopic record of diet at the time of formation [45,46]. For stable isotope analyses, cleaned and cut feathers were packed into tin capsules (aliquots: 0.70 ± 0.01 mg [mean ± s.e.]). Stable isotope analyses were conducted at the Natural Environment Research Council (NERC) Life Sciences Mass Spectrometry Facility in East Kilbride. Stable isotope ratios of carbon and nitrogen were determined by a continuous-flow mass spectrometer (Thermo Scientific [Bremen, Germany] Delta Plus XP) coupled with an elemental analyser (Elementar [Langenselbold, Germany] vario PYRO cube). To correct for instrument drift, three internal laboratory standards were analysed for every 10 samples. Stable isotope ratios are reported as $\delta$-values and expressed as ‰ according to the equation: $\delta X = [(R_{\text{sample}}/R_{\text{standard}}) - 1] \times 10^3$, where X is $^{13}$C or $^{15}$N, $R$ is the corresponding ratio $^{13}$C/$^{12}$C or $^{15}$N/$^{14}$N and $R_{\text{standard}}$ is the ratio of international references Vienna PeeDee Belemnite for carbon and atmospheric N$_2$ (air) for nitrogen. Measurement precision (standard deviation associated with replicate runs of USGS40) was ±0.1‰ for $\delta^{13}$C and ±0.2‰ for $\delta^{15}$N.

## (d) Data analysis

Analyses were conducted using R v. 3.4.4 [47]. One-way ANOVAs and *post hoc* Tukey's HSD tests were used to test for differences among reported THg concentrations (based on three feathers pooled) from previous studies of grey-headed albatrosses at South Georgia. Generalized linear mixed-effects models (GLMMs; gamma distribution and identity link function) were used to assess variation in feather THg concentrations using the 'lme4' package in R [48]. Predictor variables were trophic level ($\delta^{15}$N), foraging habitat ($\delta^{13}$C), sex (males, $n = 48$; females, $n = 30$), age (range: 12–36 years) and previous breeding outcome. Although grey-headed albatrosses are predominantly biennial breeders, with a non-breeding period lasting approximately 16 months, a minority attempt to breed annually [49]. Individuals were therefore grouped according to their breeding outcomes (successful, failed or deferred) in the 2 years prior to sampling. Individual identity was included as a random effect to account for repeated measurements. Feather $\delta^{15}$N and $\delta^{13}$C values were included in the same models as there was no evidence of collinearity (all variance inflation factors greater than 2). All possible models were ranked using the Akaike Information Criteria adjusted for small sample sizes (AIC$_C$), and models within two AIC$_C$ units of the top model (≤2 AIC$_C$) were considered equally plausible [50]. AIC$_C$ weights ($\omega_i$) were used to assess the weight of evidence in favour of a given model among the candidate set [50]. Predictor variables were standardized (i.e. subtract by mean and divide by standard deviation) to facilitate model fitting. In a second step, one-way ANOVAs with *post hoc* Tukey's HSD tests were to identify significant differences in feather THg concentrations among foraging zones. Feathers were assigned to foraging zones based on their $\delta^{13}$C values—those corresponding to foraging in the Antarctic Zone (south of the APF; less than −21.2‰), SAZ (north of the APF and south of the STF; −21.2‰ to −18.3 ‰) and the Subtropical Zone (north of the STF; greater than −18.3‰) [51].

Spearman's rank correlations were used to test for relationships between feather THg concentrations and arrival date (Julian days) of grey-headed albatrosses at South Georgia. Binomial generalized linear models (GLMs; binomial distribution and logit link function) were used to test for a relationship between feather THg concentrations and the subsequent breeding outcome (failed, $n = 55$; successful, $n = 23$). Predictor variables retained in the previous models were included as covariates, and males and females were analysed separately. Grey-headed albatrosses lay a single egg clutch with no replacement, and both parents incubate the egg. Significance was assumed at $\alpha = 0.05$ in all analyses.

## 3. Results

### (a) Temporal trends and spatial variation

THg concentrations were measured in 229 body feathers from 78 individual grey-headed albatrosses sampled in 2013/2014. Average THg concentrations of body feathers were 13.04 ± 8.03 µg g$^{-1}$ dw (range: 2.06–35.14 µg g$^{-1}$ dw), and all measurements were greater than 2 µg g$^{-1}$ dw. A total of 190 (83%) feathers had THg concentrations greater than 5 µg g$^{-1}$ dw. Feather THg concentrations measured in grey-headed albatrosses in previous studies at South Georgia, and other island groups are presented in table 1. Average feather THg concentrations for the South Georgia population in 2013 were higher than previously recorded at Campbell Island, Marion Island (Prince Edward Islands) and South Georgia. Average feather THg concentrations differed between the four previous studies at South Georgia (one-

**Table 2.** Model selection for factors explaining variation in feather total Hg concentrations ($\mu g\ g^{-1}$ dw) in grey-headed albatrosses (*T. chrysostoma*) from South Georgia, sampled in 2013/2014. The top five models are shown, and all are GLMMs with individual identity included as a random effect. $k$, number of parameters; $AIC_C$, Akaike information criterion corrected for small sample sizes; $\Delta AIC_C$ is the change in $AIC_C$ from the best-supported model and $\omega_i$ is the Akaike weight.

| covariates | | | | | | | | |
|---|---|---|---|---|---|---|---|---|
| $\delta^{13}C$ | $\delta^{15}N$ | sex | age | breeding history | $k$ | $AIC_C$ | $\Delta AIC_C$ | $\omega_i$ |
| X | X | | | | 5 | 1376.1 | 0.00 | 0.51 |
| X | X | X | | | 6 | 1378.0 | 1.90 | 0.20 |
| X | X | | X | | 6 | 1378.2 | 2.11 | 0.18 |
| X | X | X | X | | 7 | 1380.1 | 4.02 | 0.07 |
| X | X | | | X | 9 | 1382.6 | 6.58 | 0.02 |

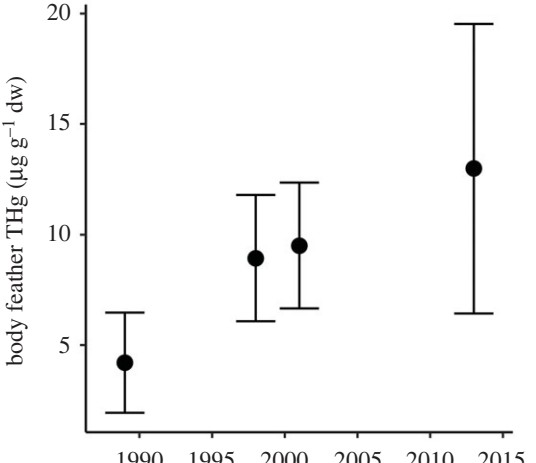

**Figure 2.** Temporal increase in mean (±s.d.) total Hg concentrations ($\mu g\ g^{-1}$ dw) of body feathers of grey-headed albatrosses (*T. chrysostoma*) from Bird Island, South Georgia. Data are based on multiple body feathers pooled per individual.

way ANOVA, $F_{3,143} = 23.6$, $p < 0.001$), and *post hoc* Tukey's HSD tests (all $p < 0.05$) confirmed that feather THg concentrations in 2013 were higher than in 1989, 1998 and 2001 (figure 2).

## (b) Variation in feather total mercury concentrations

The most parsimonious GLMM explaining variation in feather THg concentrations included the effects of $\delta^{15}N$ (estimate ± s.e., $3.37 \pm 0.36$, $p < 0.0001$) and $\delta^{13}C$ ($-1.34 \pm 0.35$, $p < 0.001$) (table 2), reflecting positive relationships with trophic level and latitude (figures 3 and 4). A similar model that also contained sex as a non-significant fixed effect ($0.75 \pm 1.63$, $p = 0.64$) had less than 2 $\Delta AIC_C$; however, this model had a greatly reduced $\omega_i$. Models including other predictor variables (age and breeding history) received less support (all greater than 2 $\Delta AIC_C$). Feather THg concentrations were significantly different between moulting zones (one-way ANOVA, $F_{2,226} = 22.41$, $p < 0.001$). *Post hoc* Tukey's HSD tests indicated that feathers associated with moulting in the AZ (mean ± s.d., $4.12 \pm 1.20\ \mu g\ g^{-1}$ dw, $n = 27$) had lower THg concentrations than those associated with the SAZ ($14.33 \pm 7.88\ \mu g\ g^{-1}$ dw, $n = 184$) or the STZ ($13.27 \pm 7.13\ \mu g\ g^{-1}$ dw, $n = 18$); means for these last two groups were not significantly different (figure 4).

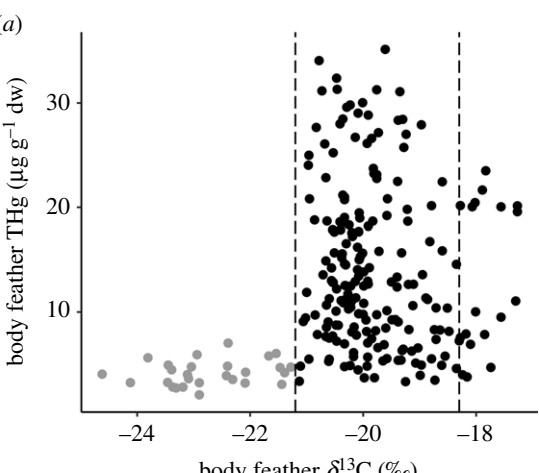

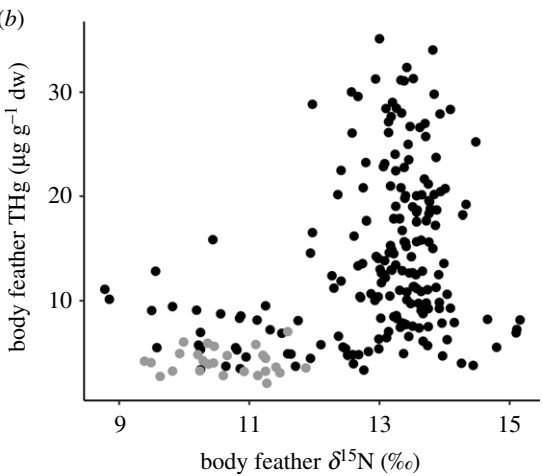

**Figure 3.** Total Hg concentrations ($\mu g\ g^{-1}$ dw) in body feathers of grey-headed albatrosses (*T. chrysostoma*) from Bird Island, South Georgia, sampled in 2013/2014, in relation to: (*a*) feather $\delta^{13}C$ values and (*b*) feather $\delta^{15}N$ values. Vertical dashed lines are $\delta^{13}C$ estimates associated with foraging at the APF ($-21.2‰$) and STF ($-18.3‰$) [51]. Body feathers with $\delta^{13}C$ values associated with foraging south of the APF are coloured grey.

## (c) Relationships between total mercury concentrations and subsequent breeding outcome

No significant relationships were found between feather THg concentrations and arrival dates of males ($r_S = 0.07$, $p = 0.40$) or females ($r_S = -0.01$, $p = 0.92$). Average THg concentrations were significantly higher in body feathers of

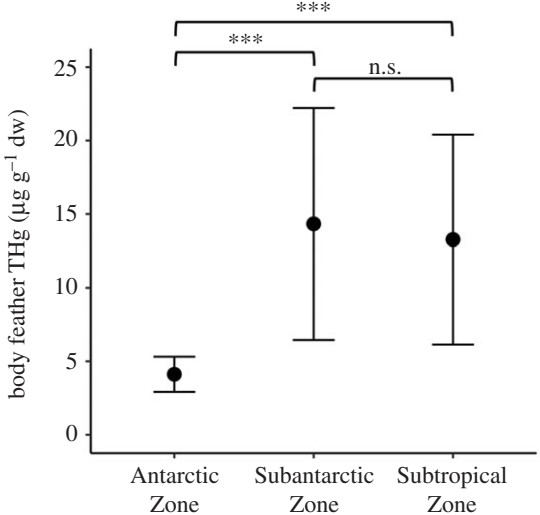

**Figure 4.** Mean (±s.d.) body feather total Hg concentrations (µg g$^{-1}$ dw) of grey-headed albatrosses (*T. chrysostoma*) from Bird Island, South Georgia, sampled in 2013/2014, in relation to moulting zones. The Antarctic Zone (AZ; south of the Antarctic Polar Front), Subantarctic Zone (SAZ; between the Antarctic Polar Front and the Subtropical Front) and Subtropical Zone (STZ; north of the Subtropical Front) are separated by $\delta^{13}$C estimates of the APF (−21.2‰) and STF (−18.3‰) [51].

males that failed to fledge a chick in the sampling year (14.96 ± 9.05 µg g$^{-1}$ dw) compared with those that were successful (11.53 ± 6.66 µg g$^{-1}$ dw; binomial GLM: $\chi^2 = 11.30$, $p < 0.001$; figure 5), but were similar among females that failed (11.81 ± 7.56 µg g$^{-1}$ dw) or were successful in fledging their chick (11.99 ± 6.04 µg g$^{-1}$ dw; $\chi^2 = 0.31$, $p = 0.58$).

## 4. Discussion

Feather analyses offer a non-lethal method to obtain information about Hg exposure during the non-breeding period, and measuring THg in albatross feathers provides information about contamination of prey and hence exposure to MeHg in the food web [19,38,52]. This study provides a detailed analysis of the underlying drivers and consequences of high feather THg concentrations in the endangered grey-headed albatross at South Georgia. Results from the present work also provide new insights into long-term changes in Hg exposure of this species and differences in exposure among breeding populations throughout the Southern Ocean.

### (a) Temporal trends in mercury exposure (1989–2014)

By comparison with previous ecotoxicological studies at South Georgia [12,14,39], the present study found a striking threefold increase in mean body feather THg concentrations of grey-headed albatrosses since the late-1980s. A similar increase has been found in other marine predators foraging in subantarctic waters. For instance, in the southern Indian Ocean, macaroni penguins (*Eudyptes chrysolophus*) and gentoo penguins (*Pygoscelis papua*) had higher feather THg concentrations in 2007 compared with the 1970s [11], and feather THg concentrations from black-browed albatross (*Thalassarche melanophris*) at the Falkland Islands have increased since 1986 [53]. Moreover, at Gough Island, Atlantic petrels (*Pterodroma incerta*), soft-plumaged petrels (*Pterodroma mollis*) and sooty albatrosses (*Phoebetria fusca*) all had higher feather THg

concentrations in 2009/2010 compared with the mid-1980s [13]. Similar trends are not observed in lower trophic level organisms (cephalopod and myctophid species) at South Georgia [54,55]; however, these samples were collected far to the south of the subantarctic and subtropical waters used by the majority of grey-headed albatrosses.

There are two plausible explanations for the temporal trend observed in our study. Firstly, grey-headed albatrosses may have shifted their diets or foraging habitats to more contaminated prey or regions. Analyses of stomach contents of chicks have revealed a major dietary shift in breeding grey-headed albatrosses at South Georgia since the mid-1990s, including a reduction in the occurrence of the seven-star flying squid (*Martialia hyadesi*) and an increase in mackerel icefish (*Champsocephalus gunnari*) [34]. However, adults may not consume the same prey that they provision to chicks and, given the difficulties in obtaining samples, the only conventional diet information for grey-headed albatrosses outside of the breeding period is for the cephalopod component [56]. Regardless, differences in stable isotope ratios of grey-headed albatross feathers from 2001 ($\delta^{15}$N, 10.48 ± 0.89‰; $\delta^{13}$C, −19.17 ± 1.12‰) contrast with the values of 12.71 ± 1.33‰ and −20.09 ± 1.30‰ in our study and provide support for the dietary shift hypothesis [12]. Moreover, the increased variance in stable isotope ratios corresponds with the higher standard deviations of THg concentrations in our study. Secondly, exposure of MeHg to grey-headed albatrosses at South Georgia may have increased. Much of the Hg that enters marine food webs originates from low-oxygen subsurface waters [57,58]. In a warming world, oxygen minimum zones are expected to increase, hence potentially enhancing methylation of Hg and its bioavailability to marine predators. Moreover, artisanal and small-scale gold mining, a major source of Hg contamination that is prevalent in South America, is increasing [4]. Rivers may deliver large amounts of Hg into the oceans [59]. Potentially, Hg in rivers flowing onto the Patagonian Shelf may be carried south in the Brazil Current and, at the confluence with the Falklands Current, be transported east in the South Atlantic Current to grey-headed albatross foraging areas [33,35,36]. Both hypotheses require further investigation, including via direct measurements of Hg in water samples or prey of grey-headed albatrosses.

### (b) Comparisons with other populations and albatross species

Average body feather THg concentrations in grey-headed albatrosses from South Georgia, sampled in 2013/2014, were higher than those reported for this species at either Campbell or Marion Island [18,39]. Non-breeding birds from these sites are oceanic foragers that predominantly target subantarctic waters [60]; however, there is considerable spatial segregation between the two populations that have been tracked during the non-breeding period—birds from Marion avoid core regions in the southwest Atlantic used by birds from South Georgia [35]. Birds from other Indian Ocean populations are likely to do the same. Accordingly, birds from all populations may use broadly the same habitat type, but our results indicate that the South Georgia population is exposed to higher Hg levels in the southwest Atlantic, potentially for the reasons mentioned above. To the best of our knowledge, there are no published data on Hg contamination for the grey-headed

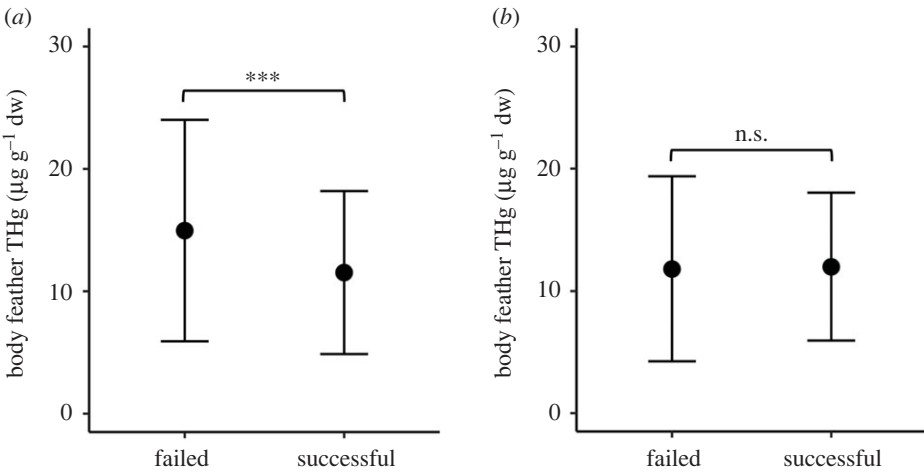

**Figure 5.** Total Hg concentrations ($\mu$g g$^{-1}$ dw) in body feathers of grey-headed albatrosses (*T. chrysostoma*) that failed to fledge a chick (failed) or successfully fledged a chick (successful) at Bird Island, South Georgia in the 2013/2014 breeding season. Males (*a*) and females (*b*) were analysed separately. Values are means and standard deviations.

albatross populations at Crozet, Kerguelen, Macquarie or the island groups off southern Chile. Phylogeny is a significant driver of Hg contamination in albatrosses, such that species in the genus *Thalassarche* tend to have lower concentrations than *Phoebetria* or *Diomedea* [18]. However, the mean feather THg concentrations in grey-headed albatrosses reported here are higher than in light-mantled albatross (*Phoebetria palpebrata*) at any site, and even the southern royal albatross (*Diomedea epomophora*) and northern royal albatross (*D. sanfordi*); indeed, they are the highest recorded for any member of the *Thalassarche* genus [18; and references therein].

## (c) Factors underlying feather total mercury concentrations

Foraging habitat, inferred from $\delta^{13}$C values, was an important driver of body feather THg concentrations. The Southern Ocean shows a latitudinal decrease in $\delta^{13}$C from subtropical to Antarctic waters, which is reflected in the tissues of albatrosses [51,61,62], and body feather THg concentrations in albatrosses appear to reflect MeHg intake during growth [18]. The threshold $\delta^{13}$C values that we used to assign moulting location to north or south of the APF and STF were derived from tracked wandering albatrosses in the Indian Ocean [51]. However, the paths of oceanographic fronts can be highly variable between years [63], and stable isotope values are therefore broadly indicative of water masses rather than latitude *per se*. Allowing for some uncertainty, our results are nevertheless indicative of lower THg concentrations in feathers of grey-headed albatrosses grown in Antarctic compared with subantarctic and subtropical waters. A near identical pattern was found in the light-mantled albatross from the Kerguelen Islands, sooty albatross from Gough Island and grey-headed albatross at Marion Island [18]. Moreover, a similar latitudinal pattern in Hg exposure has been found in a number of Southern Hemisphere seabird species, of varying trophic levels, including wandering albatross at the Crozet archipelago [26,27], as well as chicks of skuas (*Stercorarius* spp.) and multiple penguin species from the southern Indian Ocean [11,64]. Community-level studies have also documented the same trend with latitude [12,15,65]. A recent review including all 20 albatross taxa breeding in the Southern Ocean found foraging habitat ($\delta^{13}$C) to be an important driver of feather THg

concentrations, with a similar step increase north of the APF as in our study [18]. The majority of MeHg accumulated by seabirds is of mesopelagic origin, and recent work suggests that more efficient Hg methylation at depth, combined with higher vertical mixing, in subtropical compared with higher latitude waters could bring newly formed MeHg to the surface and hence increase bioavailability to seabirds [66].

In our analyses, feather THg concentrations were also positively related to $\delta^{15}$N values, which provide a proxy for trophic level. This is reflective of the biomagnification of MeHg in food webs. The relationship between $\delta^{15}$N and Hg exposure is often apparent when comparing mean values for different species within seabird communities [10,15], but is less frequently observed within a single species [8]. That it was apparent in our study population is probably because grey-headed albatrosses at South Georgia consume a wide range of prey from multiple trophic levels [34], and the variation in $\delta^{15}$N values among individuals was very high (range: 8.8–15.2‰).

A model also containing sex received less support in explaining feather THg concentrations; however, males exhibited higher feather concentrations than females. Male grey-headed albatrosses at South Georgia are heavier, with larger wing areas and higher wing loadings than females [67]. During the non-breeding period, males forage at slightly higher trophic levels and at higher latitudes compared with females [36], and tracking data show that core areas but not overall distributions were segregated to some extent during the non-breeding summer only [35,37]. No effects of age were found in the present study, which is in agreement with results from other albatross species [26–28,39], and previous studies of grey-headed albatrosses with much smaller sample sizes [14,39]. However, it should be noted that our study was cross-sectional, and it is unknown whether Hg exposure affects adult survival in grey-headed albatrosses. Hence, selective mortality of particular phenotypes cannot be excluded except by conducting a longitudinal study [32]. Stable isotope ratios of adult seabird feathers provide information about foraging ecology in the non-breeding period [36,60]. However, owing to different integration periods, it has been suggested that relationships between stable isotopes and THg concentrations may be spurious [68]; nonetheless, in albatrosses, both stable isotope ratios

and THg concentrations appear to reflect diet during feather synthesis [18; and references therein].

## (d) Fitness correlates of mercury exposure

Albatrosses are likely better adapted than other birds to Hg exposure and may tolerate higher concentrations; regardless, albatrosses are *k*-selected species and should prioritize adult survival over the immediate reproductive event. In our study, Hg exposure was clearly not sufficient to cause direct morality, or to prompt birds to defer breeding. At South Georgia, most reproductive failures occur during incubation, and consistently successful grey-headed albatrosses arrive earlier at the colony, have shorter incubation shifts and hatch larger chicks with higher growth rates compared with less successful birds [41,69]. No significant relationships between Hg exposure and arrival date were found; however, feather THg concentrations of male grey-headed albatrosses that failed in their breeding attempt were significantly higher than in those that successfully fledged a chick. Ackerman *et al.* [70] converted published Hg toxicity benchmarks in birds into blood-equivalent THg concentrations and documented negative effects with those as low as 0.2 µg g$^{-1}$ wet weight (ww) [70]. Average feather THg concentrations of failed male birds in our study were equivalent to a blood THg equivalent of 1.15 µg g$^{-1}$ ww.

Generally, health, physiology, behaviour and reproduction tend to be impacted at blood-equivalent THg concentrations of approximately 1.0 µg g$^{-1}$ ww and more substantial impairments to health and reproduction at approximately 2.0 µg g$^{-1}$ ww [70]. Male snow petrels (*Pagodroma nivea*) with higher Hg burdens are more likely to neglect eggs, and male black-legged kittiwakes (*Rissa tridactyla*) show reduced breeding success and are more likely to skip breeding (with ≤0.4 µg g$^{-1}$ ww blood THg equivalent in both cases) [24,25,70,71]. Hg exposure weakened immune function in black-footed albatrosses (*Phoebastria nigripes*) [72], though the blood-equivalent THg concentrations were far higher than birds in our study. Although there was no evidence of fitness consequences of high feather Hg contamination in a previous study of wandering albatrosses at the Crozet archipelago [26], high blood THg concentrations in the same population negatively impacted long-term breeding probability, hatching and fledging probabilities [29]. Grey-headed albatrosses are declining more steeply at South Georgia than at any other island group where there is a major population (approx. 50%); numbers are increasing at Diego Ramirez and Crozet, broadly stable at the Prince Edward Islands and declining slowly at Kerguelen [73–75]. However, the differing population trends could relate to factors other than Hg burdens, particularly the relative overlap with different fishing fleets and hence bycatch rates, which vary greatly. At South Georgia, breeding success is low, highly variable and has contributed to the negative population trends over the last 35 years [40]. Our results suggest that Hg exposure may be a contributing factor. Future work should examine birds observed as non-breeders at the colony, and also examine the risks posed by other pollutants, particularly given the increase in recovery rates per capita of marine debris (predominantly plastics) associated with albatrosses at South Georgia since the mid-1990s, and their potential role in contaminant transmission [76].

Ethics. All sampling was approved by the British Antarctic Survey Ethics Committee and carried out with permission of the Government of South Georgia and the South Sandwich Islands.

Data accessibility. The data supporting this paper are available from the Dryad Digital Repository: https://dx.doi.org/10.5061/dryad.vdncjsxsq [77].

Authors' contributions. W.F.M., P.B. and R.A.P. conceived the experimental design. R.A.P. was responsible for sample collection. W.F.M., R.A.R.M. and P.B. conducted the laboratory analyses. W.F.M. analysed the data and drafted the manuscript. All authors contributed to the writing of the manuscript, gave final approval for publication and agree to be held accountable for the work performed therein.

Competing interests. The authors declare no competing interests.

Funding. W.F.M. is supported by a NERC GW4+ Doctoral Training Partnership studentship from the Natural Environment Research Council (NERC; grant no. NE/L002434/1). Stable isotope analyses were funded by the NERC Life Sciences Mass Spectrometry Facility (grant no. EK311-12/18). The Institut Universitaire de France (IUF) is acknowledged for its support to P.B. as a Senior Member.

Acknowledgements. The authors are grateful to the many fieldworkers at Bird Island who carried out the routine monitoring and collected albatross feathers, to Maud Brault-Favrou and Carine Churlaud for their assistance with Hg analyses and to Danielle Buss for her statistical advice. Constructive comments from two anonymous reviewers substantially improved this work. Thanks are also due to the CPER (Contrat de Projet Etat-Région) and the FEDER (Fonds Européen de Développement Régional) for funding the AMA of LIENSs laboratory. This work represents a contribution to the Ecosystems component of the British Antarctic Survey Polar Science for Planet Earth Programme, funded by the Natural Environment Research Council.

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
