## [Reviewer comments · Proceedings of the Royal Society B: Biological Sciences]

Review History

RSPB-2020-1167.R0 (Original submission)

Review form: Reviewer 1

Recommendation

Reject – article is not of sufficient interest (we will consider a transfer to another journal)

Scientific importance: Is the manuscript an original and important contribution to its field?

Good

General interest: Is the paper of sufficient general interest?

Acceptable

Quality of the paper: Is the overall quality of the paper suitable?

Excellent

Is the length of the paper justified?

Yes

Should the paper be seen by a specialist statistical reviewer?

No

Do you have any concerns about statistical analyses in this paper? If so, please specify them explicitly in your report.

No

It is a condition of publication that authors make their supporting data, code and materials available - either as supplementary material or hosted in an external repository. Please rate, if applicable, the supporting data on the following criteria.

Is it accessible?

Yes

Is it clear?

Yes

Is it adequate?

Yes

Do you have any ethical concerns with this paper?

No

Comments to the Author

This paper examines factors (sex, age, breeding experience) that could influence total feather mercury concentrations in grey-headed albatrosses and uses stable isotopes to determine if foraging latitude or trophic position influence these concentrations as well. The authors compare these results to previous years and other locations where this species breeds and found that the concentrations at the colony on South Georgia Island are greater than those found at other colonies and that they have increased greatly over time. They also show that birds with greater total mercury were more likely to experience reproductive failure. I found the paper to be well written, the study to be well designed, and the analyses to be well done. I do believe that the paper would benefit from greater explanation in some sections.

The paper shows that Hg concentrations are greater at South Georgia Island compared to several other breeding colonies. How do rates of population decline or reproductive success at South Georgia compare to these other sites? Are they declining throughout the entire range, despite differences in Hg concentrations? That would suggest that Hg is not the main contributor to population decline. Do grey-headed albatross breeding on other islands forage in similar areas to those from South Georgia while molting? How far does this species travel to forage during the breeding season? Do they reach some of the higher latitude areas that are associated with greater feather Hg while those from other breeding colonies do not?

One of the key results is that failed breeders have greater total feather mercury than successful breeders, but this difference is very subtle, 2 ug/g dw. In the discussion, you need to explain how this small difference could lead to reproductive failure. Are there other studies that show small differences in feather mercury cause physiological or behavioral effects or can lead to reproductive failure?

What about the reproductive stage. Is it a failure associated with hatching or fledging the young? Date of failure may not capture this because there will be variation in when birds initiate their clutch.

You describe several negative health consequences associated with elevated Hg concentrations. How do those Hg concentrations compare to the ones you found in this study (are yours as high as some of the studies on captive birds you refer to?).

When I look at your data sheet, you have a breeding history column that includes not only if they bred successfully but also if they were seen that year. It seems that any bird you don't see is

classified as a failed breeder, but it is a failed breeder because it was absent. Being a failed breeder to me implies the bird was present, laid an egg, but then failed to fledge young. You point out that it is natural for them to usually breed every other year, so some of these failed breeders seem to just be doing what they normally should do. It seems like you should have 2 different analyses, one for years between reproductive attempts and one for initiated a clutch that failed or successful. Or, you need to clarify your data sheet with some additional explanation on the general statement page.

When during the reproductive stage does breeding failure occur? If the nest fails prior to hatching, that suggests an issue associated with the parent's exposure to Hg or perhaps body condition. But if the nest hatches but fails prior to fledging, then doesn't that suggest a less direct relationship between the parent's Hg exposure and nest failure? Date of failure may not capture this because there will be variation in when birds initiated their clutch. Does feather mercury correlate with nest initiation?

Line 158, Seems like this should say were used to test.

Line 306-308. Why is the total feather mercury showing a pattern opposite of shown by methyl mercury concentrations in ocean water?

Figure 3. The figure caption or a legend should indicate why some points are grey and others are black

Figure 4. The figure caption should indicate what the abbreviated site names on the x-axis represent.

Review form: Reviewer 2

Recommendation

Accept with minor revision (please list in comments)

Scientific importance: Is the manuscript an original and important contribution to its field?

Excellent

General interest: Is the paper of sufficient general interest?

Good

Quality of the paper: Is the overall quality of the paper suitable?

Excellent

Is the length of the paper justified?

Yes

Should the paper be seen by a specialist statistical reviewer?

No

Do you have any concerns about statistical analyses in this paper? If so, please specify them explicitly in your report.

No

It is a condition of publication that authors make their supporting data, code and materials available - either as supplementary material or hosted in an external repository. Please rate, if applicable, the supporting data on the following criteria.

Is it accessible?

Yes

Is it clear?

Yes

Is it adequate?

Yes

Do you have any ethical concerns with this paper?

No

Comments to the Author

Title: The title highlights two of the sexier findings in the article but does not truly describe the study. The temporal trend data are based on a loose comparison with other studies, and the difference in Hg levels between successful and unsuccessful birds is suggestive at best, and does not show a "consequence" of mercury exposure, which suggests causation. A more descriptive and accurate title would be "Mercury exposure in an endangered albatross: feather concentrations with respect to diet, breeding success, and past studies."

Abstract: This accurately relates the results on a clear manner.

Introduction: This section is efficient and well organized.

Methods: The procedure for combining the results from the two or more aliquots, and then the results for the three feathers per bird should be described. Were the six or more sample values simply averaged, or was weighted averaging used, or something else? Why was the number of aliquots variable between feathers. It is hard to evaluate what an SD of <10% "between runs" means, especially in the absence of understanding the previous step, in terms of what was being compared to what. Is a "run" an entire batch of samples? If these aliquots are considered duplicates, wouldn't it be better to present relative percent difference between pairs of duplicate samples from the same feather? That's what is normally judged to be "OK" if <10%. In general the QAQC section for mercury analysis is limited and unclear.

Results: The adverse effects levels of 5-40 ppm are extremely arbitrary and have little basis in fact. I agree that the authors need to pick some level, and these may be the benchmarks to use for lack of others, but they should not be perpetuated as "toxicity thresholds" as in line 197. They are based on archaic methods and speculation and are a range within which the true threshold supposedly lies. There are several articles indicating lower effects thresholds (see recent Ackerman et al. review already cited).

Discussion: In general the article is well-written and the Discussion sticks to the data. However, the overall viewpoint of the paper is rather myopically focused on albatrosses and closely related seabirds, and could benefit from bringing in some mention of the many other studies, especially effects studies, that have been done with mercury and non-albatross birds. This will broaden appeal and strengthen conclusions - as is, only two papers from non-marine birds are cited, both rather tangentially. Mercury is mercury and birds are birds, so authors should take advantage of what is known about mechanisms of reproductive failure especially. There are also numerous (hundreds) of papers on temporal trends in mercury in biota - bringing in a few of these to buttress and interpret their claim that levels are rising might also be good.

Decision letter (RSPB-2020-1167.R0)

23-Jun-2020

Dear Mr Mills:

I am writing to inform you that your manuscript RSPB-2020-1167 entitled "Mercury exposure in an endangered albatross: long-term changes and consequences for breeding success" has, in its current form, been rejected for publication in Proceedings B.

This action has been taken on the advice of referees, who have recommended that substantial revisions are necessary. With this in mind we would be happy to consider a resubmission, provided the comments of the referees are fully addressed. However please note that this is not a provisional acceptance.

Sincerely,
Dr Locke Rowe
mailto:proceedingsb@royalsociety.org

Associate Editor
Board Member: 1
Comments to Author:

Thank you for submitting your manuscript "Mercury exposure in an endangered albatross: long-term changes and consequences for breeding success" to Proceedings B. I've now received two reviews of your manuscript and reviewed the paper myself. Both reviewers were positive about your manuscript but suggested revisions that need to be addressed. Reviewer 1 has asked for better clarification and classification of failed breeding attempts, which should help pinpoint reproductive stages more strongly affected by Hg exposure and how Hg affects reproduction. Both reviewers felt the article was a bit too myopic for the readership of Proc B, and Reviewer 2 suggested ways to broaden the scope of the manuscript. Reviewer 2 also makes an interesting point about "toxicity thresholds"

Reviewer(s)' Comments to Author:
Referee: 1

Comments to the Author(s)

This paper examines factors (sex, age, breeding experience) that could influence total feather mercury concentrations in grey-headed albatrosses and uses stable isotopes to determine if foraging latitude or trophic position influence these concentrations as well. The authors compare these results to previous years and other locations where this species breeds and found that the concentrations at the colony on South Georgia Island are greater than those found at other colonies and that they have increased greatly over time. They also show that birds with greater total mercury were more likely to experience reproductive failure. I found the paper to be well

written, the study to be well designed, and the analyses to be well done. I do believe that the paper would benefit from greater explanation in some sections.

The paper shows that Hg concentrations are greater at South Georgia Island compared to several other breeding colonies. How do rates of population decline or reproductive success at South Georgia compare to these other sties? Are they declining throughout the entire range, despite differences in Hg concentrations? That would suggest that Hg is not the main contributor to population decline. Do grey-headed albatross breeding on other islands forage in similar areas to those from South Georgia while molting? How far does this species travel to forage during the breeding season? Do they reach some of the higher latitude areas that are associated with greater feather Hg while those from other breeding colonies do not?

One of the key results is that failed breeders have greater total feather mercury than successful breeders, but this difference is very subtle, 2 ug/g dw. In the discussion, you need to explain how this small difference could lead to reproductive failure. Are there other studies that show small differences in feather mercury cause physiological or behavioral effects or can lead to reproductive failure?

What about the reproductive stage. Is it a failure associated with hatching or fledging the young? Date of failure may not capture this because there will be variation in when birds initiate their clutch.

You describe several negative health consequences associated with elevated Hg concentrations. How do those Hg concentrations compare to the ones you found in this study (are yours as high as some of the studies on captive birds you refer to?).

When I look at your data sheet, you have a breeding history column that includes not only if they bred successfully but also if they were seen that year. It seems that any bird you don't see is classified as a failed breeder, but it is a failed breeder because it was absent. Being a failed breeder to me implies the bird was present, laid an egg, but then failed to fledge young. You point out that it is natural for them to usually breed every other year, so some of these failed breeders seem to just be doing what they normally should do. It seems like you should have 2 different analyses, one for years between reproductive attempts and one for initiated a clutch that failed or successful. Or, you need to clarify your data sheet with some additional explanation on the general statement page.

When during the reproductive stage does breeding failure occur? If the nest fails prior to hatching, that suggests an issue associated with the parent's exposure to Hg or perhaps body condition. But if the nest hatches but fails prior to fledging, then doesn't that suggest a less direct relationship between the parent's Hg exposure and nest failure? Date of failure may not capture this because there will be variation in when birds initiated their clutch. Does feather mercury correlate with nest initiation?

Line 158, Seems like this should say were used to test.

Line 306-308. Why is the total feather mercury showing a pattern opposite of shown by methyl mercury concentrations in ocean water?

Figure 3. The figure caption or a legend should indicate why some points are grey and others are black

Figure 4. The figure caption should indicate what the abbreviated site names on the x-axis represent.

Referee: 2

Comments to the Author(s)

Title: The title highlights two of the sexier findings in the article but does not truly describe the study. The temporal trend data are based on a loose comparison with other studies, and the difference in Hg levels between successful and unsuccessful birds is suggestive at best, and does not show a "consequence" of mercury exposure, which suggests causation. A more descriptive

and accurate title would be "Mercury exposure in an endangered albatross: feather concentrations with respect to diet, breeding success, and past studies."

Abstract: This accurately relates the results on a clear manner.

Introduction: This section is efficient and well organized.

Methods: The procedure for combining the results from the two or more aliquots, and then the results for the three feathers per bird should be described. Were the six or more sample values simply averaged, or was weighted averaging used, or something else? Why was the number of aliquots variable between feathers. It is hard to evaluate what an SD of <10% "between runs" means, especially in the absence of understanding the previous step, in terms of what was being compared to what. Is a "run" an entire batch of samples? If these aliquots are considered duplicates, wouldn't it be better to present relative percent difference between pairs of duplicate samples from the same feather? That's what is normally judged to be "OK" if <10%. In general the QAQC section for mercury analysis is limited and unclear.

Results: The adverse effects levels of 5-40 ppm are extremely arbitrary and have little basis in fact. I agree that the authors need to pick some level, and these may be the benchmarks to use for lack of others, but they should not be perpetuated as "toxicity thresholds" as in line 197. They are based on archaic methods and speculation and are a range within which the true threshold supposedly lies. There are several articles indicating lower effects thresholds (see recent Ackerman et al. review already cited).

Discussion: In general the article is well-written and the Discussion sticks to the data. However, the overall viewpoint of the paper is rather myopically focused on albatrosses and closely related seabirds, and could benefit from bringing in some mention of the many other studies, especially effects studies, that have been done with mercury and non-albatross birds. This will broaden appeal and strengthen conclusions - as is, only two papers from non-marine birds are cited, both rather tangentially. Mercury is mercury and birds are birds, so authors should take advantage of what is known about mechanisms of reproductive failure especially. There are also numerous (hundreds) of papers on temporal trends in mercury in biota - bringing in a few of these to buttress and interpret their claim that levels are rising might also be good.

Author's Response to Decision Letter for (RSPB-2020-1167.R0)

See Appendix A.

RSPB-2020-2683.R0

Review form: Reviewer 1

Recommendation

Accept as is

Scientific importance: Is the manuscript an original and important contribution to its field?

Good

General interest: Is the paper of sufficient general interest?

Good

Quality of the paper: Is the overall quality of the paper suitable?

Excellent

Is the length of the paper justified?

Yes

Should the paper be seen by a specialist statistical reviewer?

No

Do you have any concerns about statistical analyses in this paper? If so, please specify them explicitly in your report.

No

It is a condition of publication that authors make their supporting data, code and materials available - either as supplementary material or hosted in an external repository. Please rate, if applicable, the supporting data on the following criteria.

Is it accessible?

Yes

Is it clear?

Yes

Is it adequate?

Yes

Do you have any ethical concerns with this paper?

No

Comments to the Author

I think the authors did a very good job addressing my comments. There were just a couple of minor edits that should be made.

Line 67. Fix wording

Line 348. Fix wording

Line 355. Explanation of wet weight abbreviation should come here

Decision letter (RSPB-2020-2683.R0)

25-Nov-2020

Dear Mr Mills

I am pleased to inform you that your manuscript RSPB-2020-2683 entitled "Mercury exposure in an endangered seabird: long-term changes and relationships with trophic ecology and breeding success" has been accepted for publication in Proceedings B.

The referee(s) have recommended publication, but also suggest some minor revisions to your manuscript. Therefore, I invite you to respond to the referee(s)' comments and revise your manuscript. Because the schedule for publication is very tight, it is a condition of publication that you submit the revised version of your manuscript within 7 days. If you do not think you will be able to meet this date please let us know.

To revise your manuscript, log into <https://mc.manuscriptcentral.com/prsb> and enter your Author Centre, where you will find your manuscript title listed under "Manuscripts with Decisions." Under "Actions," click on "Create a Revision." Your manuscript number has been appended to denote a revision. You will be unable to make your revisions on the originally

submitted version of the manuscript. Instead, revise your manuscript and upload a new version through your Author Centre.

Sincerely,
Dr Locke Rowe
mailto: proceedingsb@royalsociety.org

Associate Editor
Board Member

Comments to Author:

Thank you for submitting your manuscript, "Mercury exposure in an endangered seabird: long-term changes and relationships with trophic ecology and breeding success" to Proceedings of the Royal Society, B. The reviewers and I have now assessed your revised manuscript and have no additional comments.

Reviewer(s)' Comments to Author:

Referee: 1

Comments to the Author(s).

I think the authors did a very good job addressing my comments. There were just a couple of minor edits that should be made.

Line 67. Fix wording

Line 348. Fix wording

Line 355. Explanation of wet weight abbreviation should come here

Decision letter (RSPB-2020-2683.R1)

27-Nov-2020

Dear Mr Mills

I am pleased to inform you that your manuscript entitled "Mercury exposure in an endangered seabird: long-term changes and relationships with trophic ecology and breeding success" has been accepted for publication in Proceedings B.

Open Access

Paper charges

Sincerely,

Appendix A

RESPONSE TO REFEREES

ASSOCIATE EDITOR BOARD MEMBER – COMMENTS TO THE AUTHOR(S)

COMMENT: Thank you for submitting your manuscript "Mercury exposure in an endangered albatross: long-term changes and consequences for breeding success" to Proceedings B. I've now received two reviews of your manuscript and reviewed the paper myself. Both reviewers were positive about your manuscript but suggested revisions that need to be addressed. Reviewer 1 has asked for better clarification and classification of failed breeding attempts, which should help pinpoint reproductive stages more strongly affected by Hg exposure and how Hg affects reproduction. Both reviewers felt the article was a bit too myopic for the readership of Proc B, and Reviewer 2 suggested ways to broaden the scope of the manuscript. Reviewer 2 also makes an interesting point about "toxicity thresholds". **REPLY:** Thanks for your comments and the opportunity to resubmit our manuscript to Proceedings of the Royal Society B: Biological Sciences. We have addressed all of the reviewers' comments by changes to the text (see detailed reply below).

REFEREE 1 – COMMENTS TO THE AUTHOR(S)

COMMENT: This paper examines factors (sex, age and breeding experience) that could influence total feather mercury concentrations in grey-headed albatrosses and uses stable isotopes to determine if foraging latitude or trophic position influence these concentrations as well. The authors compare these results to previous years and other locations where this species and breeds and found that the concentrations at the colony on South Georgia Island are greater than those found at other colonies and that they have increased greatly over time. They also show that birds with greater total mercury were more likely to experience reproductive failure. I found the paper to be well written, the study to be well designed, and the analyses to be well done. I do believe that the paper would benefit from greater explanation in some sections. **REPLY:** Thanks for the positive response. We have addressed your specific comments below.

COMMENT: The paper shows that Hg concentrations are greater at South Georgia Island compared to several other breeding colonies. How do rates of population decline or reproductive success at South Georgia compare to these other sties? Are they declining throughout the entire range, despite differences in Hg concentrations? That would suggest that Hg is not the main contributor to population decline. **REPLY:** Grey-headed albatrosses are declining more steeply at South Georgia than at any other island group where there is a major population; numbers are increasing at Diego Ramirez and Crozet, broadly stable at the Prince Edward Islands, and declining slowly at Kerguelen (Ryan *et al.* 2009 *African J. Mar. Sci.*; Robertson *et al.* 2017 *Polar Biol.*; Weimerskirch *et al.* 2018 *Polar Biol.*). However, the differing population trends could relate to factors other than Hg burdens, particularly the relative overlap with different fishing fleets and hence bycatch rates, which vary greatly. For that reason, we agree that the steep population decline at South Georgia does not necessarily relate to the high Hg burdens, although they may be a contributing factor. We now make this

clear in section 4(d) in the Discussion, and have left the text as it was at the end of the Introduction, i.e., retained the statement which indicates that Hg levels are a consideration, but does not imply a direct link with breeding success or the population decline “This final point [= the relationship between Hg and breeding outcome] is of particular interest given that breeding success in this population is low and highly variable, contributing to its long-term decline”.

COMMENT: Do grey-headed albatross breeding on other islands forage in similar areas to those from South Georgia while moulting? How far does this species travel to forage during the breeding season? Do they reach some of the higher latitude areas that are associated with greater feather Hg while those from other breeding colonies do not? REPLY: These are other good questions, which we have incorporated into section 4(b) in the discussion. Non-breeding birds from South Georgia, Marion Island and Campbell Island are all oceanic foragers that predominantly target subantarctic waters (Cherel *et al.* 2013 *Ecography*). However, there is considerable spatial segregation of birds from the two populations that have been tracked during the nonbreeding period, with those from Marion avoiding the core region in the southwest Atlantic used by nonbreeders from South Georgia (Clay *et al.* 2016 *Sci Rep*). Birds from the other populations in the Indian Ocean are likely to do the same. As such, it may be that although grey-headed albatrosses from all populations use broadly the same type of habitat, the birds from South Georgia are exposed to higher Hg levels in the southwest Atlantic, potentially because of Hg used for small-scale gold mining, as we suggested in the discussion.

COMMENT: One of the key results is that failed breeders have greater total feather mercury than successful breeders, but this difference is very subtle, 2 ug g⁻¹ dw. In the Discussion, you need to explain how this small difference could lead to reproductive failure. Are there other studies that show small differences in feather mercury cause physiological or behavioural effects or can lead to reproductive failure? REPLY: We now cite a review paper stating that negative effects of mercury in birds have been documented with blood-equivalent THg concentration as low as 0.2 ug g⁻¹ ww (Ackerman *et al.* 2016 *Sci. Total Environ.*).

COMMENT: When during the reproductive stage does breeding failure occur? If the nest fails prior to hatching, that suggests an issue associated with the parent’s exposure to Hg or perhaps body condition. But if the nest hatches but fails prior to fledging, then doesn’t that suggest a less direct relationship between the parent’s Hg exposure and nest failure? Date of failure may not capture this because there will be variation in when birds initiated their clutch. Does feather mercury correlate with nest initiation? REPLY: We now state in the Discussion that the majority of breeding failures in grey-headed albatrosses at South Georgia occur during incubation (Prince *et al.* 1994 *Ibis*). We also include correlations with arrival date at the colony, as consistently successful birds at South Georgia tend to arrive earlier (Cobley *et al.* 1998 *Ibis*).

COMMENT: You describe several negative health consequences associated with elevated Hg concentrations. How do those Hg concentrations compare to the ones you found in this study (are yours as high as some of the studies on captive birds you refer to?). REPLY: Following Ackerman *et al.* (2016 *Sci. Total Environ.*), we converted the average feather THg

concentrations of failed male birds to blood THg (ww) equivalents. Ackerman *et al.* (2016 *Sci. Total Environ.*) converted various published toxicity benchmarks in birds to blood THg (ww) equivalents, with which we now make comparisons.

COMMENT: When I look at your data sheet, you have a breeding history column that includes not only if they bred successfully but also if they were seen that year. It seems that any bird you don't see is classified as a failed breeder, but it is a failed breeder because it was absent. Being a failed breeder to me implies the bird was present, laid an egg, but then failed to fledge young. You point out that it is natural for them to usually breed every other year, so some of these failed breeders seem to just be doing what they normally should do. It seems like you should have 2 different analyses, one for years between reproductive attempts and one for initiated a clutch that failed or successful. Or, you need to clarify your data sheet with some additional explanation on the general statement page. **REPLY: We state in the Data Analysis section that "Individuals were therefore grouped according to their breeding outcomes (successful, failed or deferred) in the two years prior to sampling". We have now included a key in our data sheet to clarify.**

COMMENT: Seems like this should say were used to test [Line 158]. **REPLY: Text changed.**

COMMENT: Why is the total feather mercury showing a pattern opposite of shown by methyl mercury concentrations in ocean water? [Line 306-308]. **REPLY: The study on ocean water found very small differences with latitude, and involved sampling over a short period (e.g. March–April). It is a common in studies of seabirds with different at-sea distributions to find that Hg is lowest in species that forage in Antarctic waters, and highest in those that forage in Subtropical waters. A recent paper (published since our initial submission) using stable isotopes of Hg to demonstrated the majority of MeHg accumulated by seabirds is of mesopelagic origin and that more efficient Hg methylation at depth, combined with higher vertical mixing, in subtropical compared to higher latitude waters could bring newly formed MeHg to the surface and hence increase bioavailability to seabirds (Renedo *et al.* 2020 *Sci. Total Environ.*).**

COMMENT: The figure caption or a legend should indicate why some points are grey and others are black [Figure 3]. The figure caption should indicate what the abbreviated site names on the x-axis represent [Figure 4]. **REPLY: Figure legends corrected.**

REFEREE 2 – COMMENTS TO THE AUTHOR(S)

COMMENT: The title highlights two of the sexier findings in the article but does not truly describe the study. The temporal trend data are based on a loose comparison with other studies, and the difference in Hg levels between successful and unsuccessful birds is suggestive at best, and does not show a "consequence" of mercury exposure, which suggests causation. A more descriptive and accurate title would be "Mercury exposure in an endangered albatross: feather concentrations with respect to diet, breeding success, and past studies." [Title]. This accurately relates the results on a clear manner [Abstract]. This section is efficient and well organized

[Introduction]. **REPLY:** Thank you for the positive comments. We are happy to change the title along these lines, but would like to maintain some emphasis on the long-term changes, and also consider that “trophic ecology” might be more appropriate than “diet” given our analyses were of stable isotope ratios. Hence we would prefer to change to “*Mercury exposure in an endangered albatross: long-term changes and relationships with trophic ecology and breeding success*”.

COMMENT: The procedure for combining the results from the two or more aliquots, and then the results for the three feathers per bird should be described. Were the six or more sample values simply averaged, or was weighted averaging used, or something else? Why was the number of aliquots variable between feathers? It is hard to evaluate what an SD of <10% "between runs" means, especially in the absence of understanding the previous step, in terms of what was being compared to what. Is a "run" an entire batch of samples? If these aliquots are considered duplicates, wouldn't it be better to present relative percent difference between pairs of duplicate samples from the same feather? That's what is normally judged to be "OK" if <10%. In general the QA/QC section for mercury analysis is limited and unclear [Methods].

REPLY: This section has been clarified. Two THg measurements (“runs”) were made on each feather and we calculated the means and relative standard deviations between measurements on the same feather. A third measurement was made only if an erroneous result was found in the first two measurements. We have also added the SD to the mean recovery of the CRM to improve our presentation of the overall QA/QC protocol and results.

COMMENT: The adverse effects levels of 5-40 ppm are extremely arbitrary and have little basis in fact. I agree that the authors need to pick some level, and these may be the benchmarks to use for lack of others, but they should not be perpetuated as "toxicity thresholds" as in line 197. They are based on archaic methods and speculation and are a range within which the true threshold supposedly lies. There are several articles indicating lower effects thresholds (see recent Ackerman et al. review already cited) [Results]. **REPLY:** We agree that the “thresholds” are somewhat arbitrary and have removed those parts of the text.

COMMENT: In general the article is well-written and the Discussion sticks to the data. However, the overall viewpoint of the paper is rather myopically focused on albatrosses and closely related seabirds and could benefit from bringing in some mention of the many other studies, especially effects studies that have been done with mercury and non-albatross birds. This will broaden appeal and strengthen conclusions - as is, only two papers from non-marine birds are cited, both rather tangentially. Mercury is mercury and birds are birds, so authors should take advantage of what is known about mechanisms of reproductive failure especially. There are also numerous (hundreds) of papers on temporal trends in mercury in biota - bringing in a few of these to buttress and interpret their claim that levels are rising might also be good [Discussion]. **REPLY:** Thanks for the comment. We have tried to broaden the scope of our Discussion section, particularly in the section on ‘Fitness correlates of Hg contamination’.